# CountPath: Automating Fragment Counting in Digital Pathology

Ana Beatriz Vieira[1,2,*], Maria Valente[3,4], Tomé Albuquerque[4], Diana Montezuma[5,6], Liliana Ribeiro[5], Domingos Oliveira[5], João Monteiro[5], Sofia Gonçalves[5], Isabel M. Pinto[5], Jaime S. Cardoso[3,4], Arlindo L. Oliveira[1,2]

[1]Instituto Superior Técnico, Lisbon, Portugal
[2]INESC-ID, Lisbon, Portugal
[3]Faculdade de Engenharia da Universidade do Porto, Porto, Portugal
[4]INESC-TEC, Porto, Portugal
[5]IMP Diagnostics, Porto, Portugal
[6]Cancer Biology and Epigenetics Group, Research Center of IPO Porto (CI-IPOP) / RISE@CI-IPOP (Health Research Network), Portuguese Oncology Institute of Porto / Porto Comprehensive Cancer Centre (Porto.CCC), Porto, Portugal

*Abstract*—Quality control of medical images is a critical component of digital pathology, ensuring that diagnostic images meet required standards. A pre-analytical task within this process is the verification of the number of specimen fragments, a process that ensures that the number of fragments on a slide matches the number documented in the macroscopic report. This step is important to ensure that the slides contain the appropriate diagnostic material from the grossing process, thereby guaranteeing the accuracy of subsequent microscopic examination and diagnosis. Traditionally, this assessment is performed manually, requiring significant time and effort while being subject to significant variability due to its subjective nature. To address these challenges, this study explores an automated approach to fragment counting using the YOLOv11 and Vision Transformer models. Our results demonstrate that the automated system achieves a level of performance comparable or even superior to that of experts, offering a reliable and efficient alternative to manual counting. Additionally, we present findings on interobserver variability, showing that the automated approach achieves an accuracy of 90.1%, surpassing the range observed among experts (82–88%). This result further supports its suitability for integration into routine pathology workflows.

*Index Terms*—Counting, detection, digital pathology, fragments, interobserver variability, ViT, YOLO.

## I. INTRODUCTION

Digital Pathology has transformed medical diagnostics by automating image analysis, improving efficiency, and streamlining pathology workflows [1]. However, the adoption of these technologies has introduced new challenges in quality control, namely in ensuring that digital slides accurately reflect the received biological material. One key pre-analytical task is detecting and counting tissue fragments on pathology slides, crucial for preventing material loss and contamination. A fragment is defined as an distinct and visually separable piece of tissue present on a histological slide, typically resulting from the sectioning of a larger specimen during the grossing and embedding stages of sample preparation. Figure 1 illustrates

---

*Corresponding author: anabeatrizvieira@tecnico.ulisboa.pt

the fragment lifecycle, from grossing to whole-slide image (WSI) acquisition, where the fragments become visible.

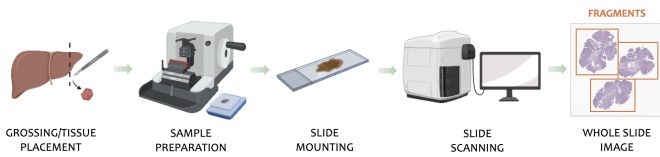

Fig. 1. Overview of the fragment lifecycle in digital pathology. Tissue fragments originate during the grossing step, where specimens are sectioned into smaller pieces. These are embedded, sliced, and mounted onto glass slides during sample preparation. After staining, the slides are scanned to produce WSIs, where three distinct fragments are visible.

Traditionally, biomedical scientists perform this task manually by comparing histology slides with macroscopic reports, a process that, while effective, is time-consuming and prone to human error, especially with high slide volumes. Accurate fragment detection and quantification are essential to verifying that the tissue analyzed corresponds precisely to what was received. Fragment variability in size, shape, and arrangement can complicate counting, and discrepancies between the observed and reported number of fragments trigger quality control reviews. These reviews help identify potential errors such as specimen misprocessing, tissue loss, or documentation issues, ensuring corrective actions are taken to maintain diagnostic accuracy.

Subjectivity is inherent in manual counting, as interpretations of fragment boundaries can vary between observers. For example, a fragment breaking during processing may prompt one observer to review the paraffin block, while another may consider the break insignificant. Whereas such variability cannot be eliminated, automated approaches can help standardize assessments and reduce inconsistencies. Currently, manual fragment counting is performed twice daily at IMP Diagnostics, taking 4–5 hours to process 1500–1600 slides. About 20–30 cases per day require review, with up to five

needing corrective procedures like re-cutting. Although the percentage of cases needing review and correction is relatively low, this is relevant to maintaining a high-quality standard. This type of quality check is important across pathology laboratories, where accuracy and consistency directly impact patient outcomes.

Automated quality control in digital pathology is increasingly recognized as essential [2,3]. However, specific tasks like fragment quantification remain largely underexplored. To the best of our knowledge, the only prior work addressing this is by Albuquerque et al. [4], who developed an automated system for detecting fragments in colorectal biopsy WSIs using both conventional machine learning and deep learning models, including YOLOv5 and Faster R-CNN. Their system improved accuracy and reduced manual workload by identifying discrepancies between slides and macroscopic reports. However, their approach was limited in key aspects. First, it focused exclusively on colorectal biopsies, leaving its generalizability to other tissue types untested. Second, while both fragments and sets were detected, the total number of fragments per image was not explicitly calculated. Third, the validation was performed on a relatively narrow dataset, with no systematic comparison against expert manual counts, limiting conclusions about interobserver variability and clinical robustness. Building on this foundation, we propose an advanced automated system for fragment detection and counting in digital pathology. Our approach reduce workload and improves accuracy and diagnostic reliability across diverse organ types. We validate its performance against manual counts from seven experts.

## II. METHODS

This section describes the methodology, including dataset details, detection and counting methods for fragments and sets in WSIs, and a rejection option to request manual revision for unreliable predictions. Finally, it describes the performed interobserver variability analysis used to assess consistency with human annotations, providing insights into reliability.

### A. Dataset

The dataset comprises 3,253 WSIs from the IMP Diagnostics archive, digitized using three Leica GT450 WSI scanners at 40× magnification. This study specifically uses $1024 \times 1024$ thumbnails (low-resolution) images representations embedded within the pyramid-encoded WSI files, providing an efficient overview of the tissue layout [5]. The dataset spans various organ types, including gastric (788), prostate (263), colorectal (1697), cervix (114) and other organs (391). Each histology slide could contain one set of different fragments (small pieces of tissue from biopsy or surgery procedures) or repeated sets of the same fragments, as shown in Figure 2. These repeated sets are typically arranged to ensure that the tissue is adequately represented across multiple sections, allowing for better analysis.

Each case was evaluated by at least one pathologist and one biomedical scientist, and the number of fragments was compared with the original macroscopic report. Discrepancies

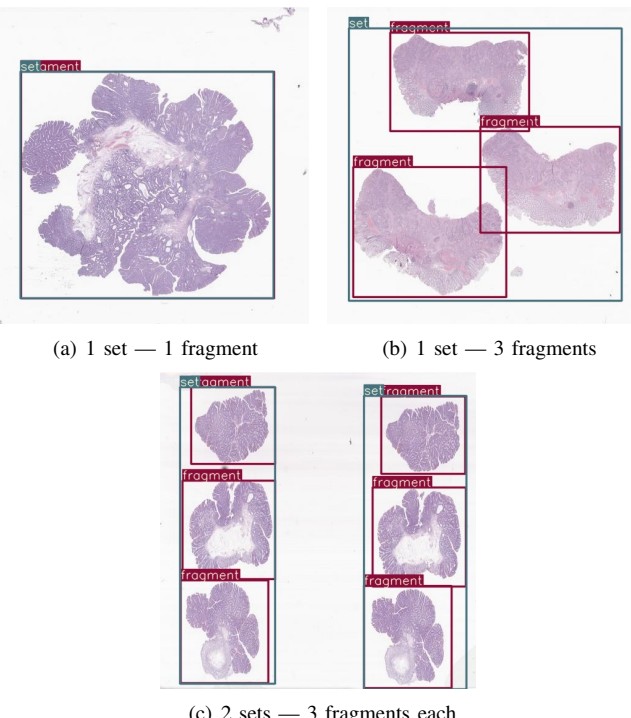

(a) 1 set — 1 fragment     (b) 1 set — 3 fragments

(c) 2 sets — 3 fragments each

Fig. 2. Representation of different tissue configurations in images. (a) One set containing a single fragment. (b) One set with three distinct fragments. (c) Two repeated sets, each containing three fragments of identical size and morphology.

were resolved through discussion between the evaluators or by consulting a third colleague. The dataset includes two types of annotations: spatial annotations, which specify bounding boxes, and numeric annotations, which indicate the number of fragments per set. The dataset contains two different types of annotations: spacial annotations (that are further divided into fragments (17,762 annotations) and sets (5,630 annotations)) and numeric annotations (701), which indicate the number of fragments per set. Notably, all images with spatial annotations also have numeric annotations, ensuring that the entire dataset is numerically annotated. Cases with more than 9 fragments per set are labeled as class 10, reflecting the laboratory's decision that an exact count is unnecessary for cases when the number of sets exceeds this value. The distribution of cases per class is as follow: class 1: 752, class 2: 1037, class 3: 511, class 4: 307, class 5: 188, class 6: 131, class 7: 73, class 8: 58, class 9: 35, class 10: 161.

### B. Detection and Counting Models

We explored two complementary methods for counting fragments and sets in histopathology images: detection and classification. The methodology includes first detecting bounding box for each fragment and for each set. Then, fragment counts are derived from model predictions and refined through post-processing rules that prevent overlapping sets and ensure fragments are counted only within defined sets. The final count is obtained by dividing the number of fragments by the number of sets. When this ratio is not an integer, indicating a

discrepancy, two strategies could be applied: (a) rounding the value to the nearest whole number or (b) flagging the sample for manual review. In contrast, the classification approach directly assigns images to predefined categories based on the number of fragments per set.

For detecting fragments and sets, we selected a state-of-the-art object detection model, YOLOv11, a convolutional neural network that builds on previous versions of the YOLO family. YOLOv11 introduces several enhancements, including improved feature extraction, advanced attention mechanisms, and optimized training strategies, leading to higher detection accuracy, faster inference, and better efficiency compared to earlier versions [6]. Pre-trained on the MS COCO dataset, the model is fine-tuned on the target dataset, with input images annotated with bounding boxes and classes. For fragments and sets counting, we used the Vision Transformer (ViT) as a classifier. ViT processes images by dividing them into patch-based tokens using self-attention mechanisms, similar to the ones used in natural language processing NLP [7]. Pre-trained on ImageNet and fine-tuned on our dataset, ViT classifies images into predefined categories that corresponds to the fragment count.

We also implemented a two-stage approach, referred to as CountPath, which combines YOLOv11 and ViT to improve counting accuracy. In the first stage, both models are used to detect and count sets, with the ViT model imposing an upper bound on the number of sets detected by YOLOv11 to mitigate potential over-counting errors. The set predictions are then refined through post-processing, which removes overlapping detections and filters out low-confidence predictions, ensuring that only reliable sets are retained. Each validated set is subsequently cropped from the original image, producing one image per set. In the second stage, a separate YOLOv11 model processes each cropped image to detect the fragments within the corresponding set. Finally, a post-processing consistency check is applied across all cropped images derived from the same original image. Figure 3 shows the mentioned methods.

### C. Rejection Option

In automated image analysis, particularly for complex medical imagery like histopathology slides, ensuring accurate classification is crucial for reliable downstream applications. While high-confidence predictions are ideal, machine learning models often produce varying confidence levels, which can lead to misinterpretations in precision-critical applications. To address this challenge, introducing a "rejection option" has emerged as a valuable strategy for filtering out samples that do not have a reliable classification. This approach acknowledges that some samples may be inherently ambiguous, leading to misclassifications. By rejecting these cases, the model's overall reliability improves. In our YOLOv11-based fragment and set detection, we implemented a simple integrity criterion: samples were rejected if the calculated fragment-to-set ratio was not an integer, indicating potential classification errors. In the CountPath approach, the rejection option is applied when the fragment counts in all cropped images from the same

original image are inconsistent, not ensuring reliability in the final count.

### D. Interobserver Variability

We selected a subset of 100 samples from the test set and asked a panel of seven biomedical scientists, with experience ranging from less than one year to over five years, to record the final score for each image. Given the inherent subjectivity in fragment counting, we assessed interobserver variability, which quantifies differences in counting among observers [8]. To measure agreement, we computed the intraclass correlation coefficient (ICC), a standard metric for assessing reliability within a group of raters [9]. Additionally, we calculated Fleiss' Kappa, an extension of Cohen's Kappa, to evaluate inter-rater reliability or agreement for multiple raters [8], [10]. Higher values of these metrics indicate stronger agreement [11].

## III. EXPERIMENTS

In this section, we present the dataset, preprocessing and post-processing steps, along with a detailed description of the hyperparameters and training configurations used for the models. All experiments were conducted on a PowerEdge C41402 server with an NVIDIA 32GB Tesla V100S.

### A. Datasets and Preprocessing

The dataset was divided into train, validation, and test subsets, containing 2,053, 499, and 701 images, respectively. Each image was resized to either $512 \times 512$ or $224 \times 224$ pixels, depending on the model architecture. After set detection, a new dataset was created by cropping these regions from the original images, resizing them to the model's input size, and applying white padding. All images were normalized from their original pixel values (0–255) to a 0–1 range. Furthermore, to align with the pre-trained ViT network, the images were normalized using the specified mean and standard deviation for 224×224 images. Given the relatively limited dataset size, various data augmentation techniques were applied during training, further detailed below, to increase the robustness of the learning process.

### B. Post-processing

The post-processing stage consists of two phases: set counting and fragment counting, each ensuring consistency and reliability in detection results. In set counting, the number of sets predicted by the ViT model serves as an upper bound, constraining YOLOv11 detections. Redundant bounding boxes are removed and remaining detections are sorted by confidence. The lower-confidence boxes are removed until the number of detections aligns with ViT's output, ensuring consistency. Each set is then used to generate a cropped region. As a result, the number of cropped images corresponds to the sets identified in the original image, ensuring each set is isolated for further processing in the fragment detection stage.

In fragment counting, the process ensures consistency by verifying whether all cropped images derived from the same original image contain the same number of fragments. The

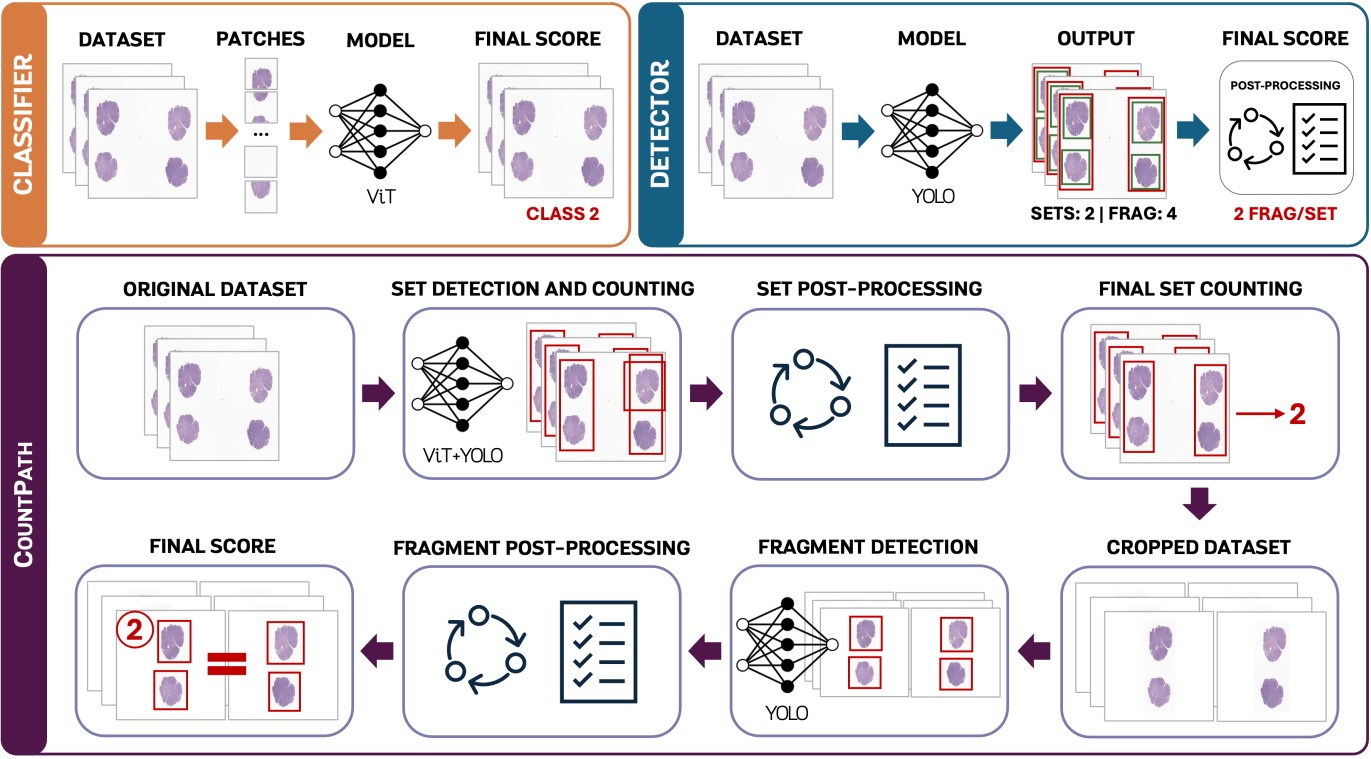

Fig. 3. **Overview of the three evaluated approaches for fragment detection and counting.** (a) The direct-counting approach uses a ViT-B/32 model to extract patches and predict the number of fragments per set. (b) The detection approach employs YOLOv11-X to identify fragments and sets, followed by post-processing to compute the final score. (c) The proposed CountPath pipeline first detects sets with YOLOv11-X and uses ViT-B/32 for set counting with over-counting control. After post-processing and cropping, a second YOLOv11-X detects fragments, and consistency is enforced across cropped regions.

count is considered valid only if all corresponding cropped regions yield the same result. In cases of discrepancy, the final score is determined according to one of two options:

- **With rejection:** the sample is flagged for manual review and excluded from evaluation;
- **Without rejection:** the most frequent fragment count is selected (majority voting). In case of a tie, the lowest among the most frequent values is used as the final score.

### C. Implementation Details

We conducted all experiments using YOLOv11 and ViT-Base, implemented in PyTorch. Prior to selecting YOLOv11-X for final evaluation, we conducted preliminary experiments with earlier YOLO versions and alternative YOLOv11 variants, including YOLO-N and YOLO-M. We also explored various training configurations by tuning hyperparameters such as learning rate, data augmentation techniques, and optimizer settings. Nonetheless, the best performance was consistently achieved using the default settings of the Ultralytics implementation of YOLOv11-X [12], which we therefore adopted. YOLOv11-X was trained for 200 epochs with a batch size of 32, using stochastic gradient descent (SGD) with an initial learning rate of $10^{-2}$, which was reduced to $10^{-3}$, and a weight decay of $5 \times 10^{-4}$. Given the limited dataset size, we applied several data augmentation techniques, including vertical and horizontal flipping, translation, HSV adjustments,

and mosaic augmentation. ViT-Base/32 was also trained for 200 epochs with a batch size of 32, using AdamW with a learning rate of $10^{-5}$ and a weight decay of $5 \times 10^{-2}$. To enhance generalization, we applied horizontal and vertical flipping as data augmentation strategies.

## IV. RESULTS AND DISCUSSION

This section presents the experimental results, offering a quantitative comparison of the ViT, YOLOv11, the baseline models LSM-YOLO and RT-DETR, and the proposed Count-Path method for fragment and set counting. Additionally, we analyze interobserver variability in fragment counting and compare it to the best-performing automated model.

### A. Quantitative Analysis

We trained several object detection models – YOLOv11-X, and two baseline architectures, RT-DETR-L [13], and LSM-YOLO [14] – to detect fragments and sets. All models were evaluated on 701 samples, with and without a rejection option. The rejection criterion was based on the ratio of detected fragments to sets: if this ration did not yield an integer, the sample was rejected. The final score was computed as the ratio of detected fragments to sets, rounded to the nearest integer when the rejection option was not applied. In parallel, we implemented a ViT-B/32 model for counting fragments and sets and compared its results against the object detection models. This model serves as a global, classification-based,

alternative to the detection-based approaches. Additionally, we developed the CountPath method that combines YOLOv11-X and ViT-B/32 architectures to count sets and fragments. In this case, the rejection option is applied when the fragment counts in all cropped images from the same original image are inconsistent, ensuring reliability in the final count. A comparative analysis of these methods is presented in Table I.

TABLE I
PERFORMANCE EVALUATION OF FRAGMENT COUNTING USING VIT-B/32, YOLOV11-X, THE BASELINE MODELS LSM-YOLO AND RT-DETR-L, AND THE PROPOSED COUNTPATH METHOD. RESULTS ARE PRESENTED BOTH WITH AND WITHOUT REJECTION CRITERIA, WHERE THE REJECTION PERCENTAGE INDICATES THE PROPORTION OF CASES EXCLUDED. FOR THE REJECTED CASES, THE EVALUATION METRICS ARE COMPUTED WITHOUT THESE SAMPLES. PRECISION, RECALL, AND F1-SCORE ARE WEIGHTED TO ACCOUNT FOR CLASS IMBALANCES.

| Model | Rejection | MAE | Accuracy | Precision | Recall | F1-Score |
|---|---|---|---|---|---|---|
| YOLOv5-N [4] | No | 0.246 | 83.1% | 83.2% | 83.1% | 83.0% |
| ViT-B/32 | No | 0.291 | 78.9% | 78.0% | 78.9% | 78.2% |
| YOLOv11-X | No | 0.136 | 90.6% | 90.9% | 90.6% | 90.5% |
| | Yes (4.14%) | 0.116 | 92.1% | 92.4% | 92.1% | 92.1% |
| LSM-YOLO | No | 0.158 | 88.4% | 88.9% | 88.4% | 88.4% |
| | Yes (3.42%) | 0.148 | 89.4% | 90.0% | 89.4% | 89.4% |
| RT-DETR-L | No | 0.126 | 90.0% | 90.3% | 90.0% | 90.0% |
| | Yes (5.85%) | 0.102 | 92.3% | 92.4% | 92.3% | 92.3% |
| CountPath | No | **0.126** | **91.9%** | **91.8%** | **91.9%** | **91.7%** |
| | Yes (3.99%) | **0.091** | **94.4%** | **94.4%** | **94.4%** | **94.3%** |

The YOLOv11-X model achieved an overall accuracy of 90.6% when classifying all 701 samples without rejection. With 4.14% of the samples rejected, the accuracy increased to 92.1%, leaving 672 samples classified and 29 rejected. The rejection criterion, whether the fragments-to-sets ratio was an integer, proved effective in filtering ambiguous samples, modestly improving performance.

The ViT-B/32 model achieved 78.9% accuracy, which is lower than all detection-based approaches. This can be partly attributed to the ambiguity in defining fragments and the differences in training data: YOLOv11-X was trained on bounding box annotations, while ViT relied on image-level labels. However, the ViT-B/32 model exhibited outstanding performance in set counting, achieving 99.3% precision and a 99.6% F1-Score. This result underscores the effectiveness of ViT and attention mechanisms in accurately identifying sets, where the definition is clear and well-defined.

The baseline models RT-DETR-L and LSM-YOLO showed comparable performance to YOLOv11-X. RT-DETR-L achieved 90.0% accuracy without rejection and 92.3% accuracy when 5.85% of the samples were excluded. LSM-YOLO performed sligthly worse, reaching 88.4% accuracy without rejection and 89.4% with 3.42% rejection.

A qualitative analysis of the errors made by each model revealed important differences in their failure modes. The RT-DETR-L model was particularly sensitive to noise, with most counting errors resulting from the incorrect detection of fragments frequently detecting artifacts as fragments. Additionally,

this architecture frequently failed to detect repeated sets, leading to inconsistencies in set counting. In contrast, both YOLOv11 and LSM-YOLO demonstrated greater robustness to artifacts, with most of their errors arising from the under-detection of sets rather than fragments. These observations suggest that, although all models exhibit specific limitations, YOLO-based approaches are generally more resilient to morphological variability and image noise, making them better suited for scenarios with high levels of artifacts.

Motivated by these complementary strengths and weaknesses, we developed the CountPath method that integrates YOLOv11 for fragment detection with a Vision Transformer (ViT) for set counting. CountPath outperformed all models, achieving an accuracy of 91.9% without rejection, which further improved to 94.4% when 3.99% of the samples were excluded. This result demonstrates that the rejection strategy effectively enhances reliability by filtering out ambiguous cases. These results underscore the advantages of integrating both architectures in a hybrid approach, leveraging the strengths of YOLOv11-X for precise fragment detection and ViT for robust set counting.

Given that the best results were achieved with the CountPath method, we conducted a detailed performance evaluation stratified by organ type. For this analysis, we divided the test set into five groups: gastric (394 samples), prostate (43 samples), colorectal (200 samples), cervix (12 samples), and others (52 samples). These groups were chosen due to their substantial representation in the dataset. The results by organ type are represented in Table II.

TABLE II
RESULTS ACHIEVED WITH THE PROPOSED COUNTPATH METHOD BY ORGAN TYPE.

| Organ | Rejection | Samples | MAE | Accuracy | Precision | Recall | F1-Score |
|---|---|---|---|---|---|---|---|
| Gastric | No | 394 | 0.086 | 93.4% | 93.4% | 93.4% | 93.2% |
| | Yes (4.06%) | 378 | 0.061 | 95.8% | 95.8% | 95.8% | 95.6% |
| Prostate | No | 43 | 0.070 | 93.0% | 95.3% | 93.0% | 94.0% |
| | Yes (4.65%) | 39 | 0.049 | 95.1% | 95.3% | 95.1% | 94.9% |
| Colorectal | No | 200 | 0.215 | 89.0% | 89.6% | 89.0% | 88.9% |
| | Yes (3.0%) | 194 | 0.165 | 91.2% | 91.6% | 91.2% | 91.2% |
| Cervix | No | 12 | 0.250 | 91.7% | 93.1% | 91.7% | 87.9% |
| | Yes (16.7%) | 10 | 0.000 | 100% | 100% | 100% | 100% |
| Others | No | 52 | 0.115 | 90.4% | 92.2% | 90.4% | 90.8% |
| | Yes (3.85%) | 50 | 0.080 | 94.0% | 96.2% | 94.0% | 94.7% |

The results indicate that the model's performance in counting fragments is consistent across different organ types. The highest accuracy is observed for the gastric sample set, which represents the largest group in the test set. In the next section we evaluate the methodology under a domain generalization scenario.

### B. Domain Generalization

Since samples can have distinct sizes, shapes, and characteristics, it is crucial to ensure that the model generalizes accurately across domains, especially in clinical settings where samples are diverse in nature, some of which may be new

to the model. Although we applied techniques like data augmentation and model regularization to improve in-domain generalization, evaluating how well the model counts fragments across samples with different characteristics, helps in the assessment of the model robustness to domain variations. To this end, we conducted a series of experiments based on previously established organ groups. We trained our method while systematically excluding each group and subsequently assessed its performance across all 701 samples. Table III presents the results when each organ is excluded from training.

TABLE III
RESULTS BY THE PROPOSED COUNTPATH METHOD BY ORGAN TYPE, WITH THE MODEL TRAINED WITHOUT EACH OF THESE ORGANS. THE TRAINING DATASET SIZES VARY DEPENDING ON THE ORGAN EXCLUDED DURING MODEL TRAINING: WHEN TRAINED WITHOUT **GASTRIC** SAMPLES, THE DATASET COMPRISES 2158 SAMPLES; WITHOUT **PROSTATE** SAMPLES, IT COMPRISES 2332 SAMPLES; WITHOUT **COLORECTAL** SAMPLES, IT COMPRISES 1055 SAMPLES; WITHOUT **CERVIX** SAMPLES, IT COMPRISES 2450 SAMPLES; AND WITHOUT **OTHERS** SAMPLES, IT COMPRISES 2213 SAMPLES.

| Organ | Rejection | Samples | MAE | Accuracy | Precision | Recall | F1-Score |
|---|---|---|---|---|---|---|---|
| Gastric | No | 394 | 0.081 | 93.7% | 93.7% | 93.7% | 93.5% |
| | Yes (4.57%) | 376 | 0.056 | 96.3% | 96.3% | 96.3% | 96.3% |
| Prostate | No | 43 | 0.395 | 76.7% | 93.3% | 76.7% | 82.1% |
| | Yes (4.65%) | 41 | 0.390 | 78.0% | 95.1% | 78.0% | 83.5% |
| Colorectal | No | 200 | 0.220 | 86.0% | 86.5% | 86.0% | 85.5% |
| | Yes (5.0%) | 190 | 0.179 | 87.9% | 88.4% | 87.9% | 87.5% |
| Cervix | No | 12 | 0.083 | 91.7% | 95.8% | 91.7% | 88.9% |
| | Yes (0.0%) | 12 | 0.083 | 91.7% | 95.8% | 91.7% | 88.9% |
| Others | No | 52 | 0.154 | 90.4% | 91.2% | 90.4% | 90.2% |
| | Yes (3.85%) | 50 | 0.080 | 94.0% | 96.2% | 94.0% | 94.7% |

The results indicate that the model is generally capable of detecting and counting fragments in samples from a given organ, even when the model has not been trained on that specific organ type. By removing each of the organs from the training set, we are significantly reducing the dataset, which translates into results slightly worse those presented above. However, the lowest performance is observed for prostate samples, possibly likely to the distinct morphological characteristics of prostatic fragments (vs. digestive samples, which account for the majority of the dataset).

A more detailed analysis of the results reveals that excluding of specific organ groups from the dataset has a direct impact on the dataset size, which, in turn, influences the model's performance. Below, we detail the outcomes of training experiences conducted without one of the organ groups:

- **Exclusion of gastric samples:** Excluding gastric samples from the training dataset results in stable model performance, indicating that these samples contribute with minimal additional information to the training process. This outcome can be attributed to the morphological similarity of gastric fragments to other samples in the dataset (namely colon samples, which account for the majority of cases), which minimizes their influence on detection and counting tasks.

- **Exclusion of prostate samples:** When prostate samples are excluded, the model shows a reduced ability to detect and count this specific organ type. However, its performance on the remaining samples remains consistent. This decrease in performance is likely due to the distinct characteristics of prostate fragments compared to other organ types in the dataset.

- **Exclusion of colorectal samples:** Removing colorectal samples, which constitute a significant proportion of the dataset (1497 samples), leads to a decrease in the model's counting performance. This decline can be attributed to the reduced size of the training set (which shrinks from 2552 to 1055 samples), resulting in less representative data for training.

- **Exclusion of cervix and "others" samples:** Excluding cervix or "others" samples has minimal impact on overall model performance metrics, primarily due to the small size of these subsets. For cervix samples, the small number of cases in the test set (12) is too limited to draw meaningful conclusions about their specific impact on the model's performance. Similarly, the exclusion of "others" samples results in negligible changes, likely because these samples are less distinct or closely resemble other groups within the dataset. These findings highlight the limited influence of subsets with very few cases on the overall performance metrics.

We can conclude that the model's strong generalization across domains shows its capability to effectively handle diverse and complex samples without significant accuracy loss.

### C. Interobserver Variability Analysis

As mentioned, a group of seven observers was asked to annotate a subset of 100 images from the original test set. To assess the agreement between the observers, we calculated both the ICC and Fleiss' Kappa, two commonly used metrics for measuring inter-rater reliability. The ICC was 0.814, and Fleiss' Kappa was 0.74, indicating substantial agreement between the evaluators [15]. Given this, we compared the performance of the automated method with the performance of the manual counting performed by these observers. It is important to note that the subset, although randomly selected, contains a high percentage of complex images that caused uncertainty or were difficult for the models to detect accurately in previous tests. The results from both the observers and the proposed method are presented in Table IV.

TABLE IV
COMPARISON BETWEEN COUNTPATH AND MANUAL COUNT BY 7 OBSERVERS ON 100 TEST SAMPLES. METRICS ARE WEIGHTED TO ACCOUNT FOR CLASS IMBALANCE.

| Method | MAE | Accuracy | Precision | Recall | F1-Score |
|---|---|---|---|---|---|
| Observers | 0.210±0.02 | 84.9%±2.0 | 77.9%±8.0 | 72.3%±6.0 | 73.5%±7.0 |
| CountPath | 0.099 | 90.1% | 90.1% | 90.1% | 89.9% |

The results show that our method outperforms the manual counting by observers in multiple evaluation metrics. The

automated approach achieves a lower Mean Absolute Error (MAE) reflecting a strong correlation between predictions and ground truth values. Additionally, it achieves higher precision, recall, and F1-score, demonstrating improved reliability in detecting and counting fragments. With an accuracy of 90.1%, surpassing the observers' range (82–88%), the automated approach proves to be a reliable alternative.

## V. Conclusion

This work addressed the challenges of manual fragment counting in digital pathology by introducing CountPath, a method that integrates YOLOv11 and Vision Transformer (ViT) architectures. CountPath was designed to enhance accuracy and reduce interobserver variability in this critical quality control task.

We systematically evaluated several methods for fragment and set counting, including the ViT-B/32 classifier, the YOLOv11-X detector, and two baseline models: LSM-YOLO and RT-DETR. While all methods demonstrated competitive performance, YOLOv11-X consistently outperformed the baselines across most metrics, particularly when combined with a rejection strategy to filter ambiguous predictions. Building upon this backbone, the CountPath method combines ViT-B/32 for set-level classification with YOLOv11-X for fragment detection, achieving the highest overall accuracy. This design leverages both local detection precision and global contextual reasoning, yielding a robust and scalable solution.

To assess robustness in real-world conditions, we evaluated the method's ability to generalize across domains by systematically excluding one organ type during training and testing on the full dataset. The results showed strong generalization performance, with only modest drops in accuracy, even when the excluded organ was morphologically distinct. Performance remained particularly stable for gastric, cervix, and "other" categories, while a significant decrease was observed for prostate samples, likely due to their unique structural characteristics. These findings highlight the method's ability to handle diverse and previously unseen tissue types, supporting its suitability for deployment in heterogeneous clinical settings.

Compared to manual counting performed by seven experts, the proposed CountPath method achieved an accuracy of 90.1%, surpassing the experts' average of 84.9%. This result demonstrates its potential to improve consistency and reliability in fragment counting tasks.

Our findings demonstrate that deep learning-based approaches, particularly the new method proposed, can improve pathology workflows by increasing accuracy, minimizing variability, and reducing the risk of diagnostic inconsistencies. By leveraging the strengths of both object detection and transformer-based models, our approach offers a robust and scalable solution for automated quality control in digital pathology.

## Acknowledgment

This work was supported by national funds through Fundação para a Ciência e a Tecnologia, I.P. (FCT) under projects UID/50021/2025 and UID/PRR/50021/2025, DIXCover and Center for Responsible AI (C628696807-00454142). In addition, IMP Diagnostics also provided funding support for this study. We want to thank the biomedical scientists that also contributed to the interobserver variability study and/or data collection (Joana Guimarães, BSc; Mara Guedes, BSc; Cátia Gonçalves, BSc; Filipa Rebolo, BSc; Joana Ferreira, BSc; Vânia Leal, BSc).

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
