# OpenReview forum: "CountPath: Automating Fragment Counting in Digital Pathology"
_IEEE.org/EMBS/BHI/2025/Conference — BHI 2025_

### Official Review · Reviewer_9Fg3 · 2025-07-11
**CountPath: Automating Fragment Counting in Digital Pathology**

**Confidence:** 5
**Clarity Of Writing:** good
**Clinical Significance:** fair
**Methodological Novelty:** fair
**Overall Rating:** 4
**Final Rating:** 6

**Experiments And Results:**

good

**Questions For The Authors:**

Explain what a fragment is in introduction may be with a sample figure. It is given too late in the methodology section.

In introduction, last paragraph, authors claim several aspects are ignored while they only list two.

Rather than using the page space for Figure 1,  a figure could help readers understand the three separate approaches defined in section II.B.

In the introduction section the shortcomings of Albuquerque et al[2] are listed but in the experimental results section the proposed method is compared against other baseline models [10,4]. Why not test it against [2]?

In Section IV. A last paragraph, the authors say  the highest accuracy is obtained with gastric images which is normal if they most probably have a higher percentage in the training dataset. If the authors would like to make a fair comparison they have to augment the dataset so that different cancer types have similar percentages in the training dataset.

**Strengths:**

The model is more robust compared to the baseline models they tested.

**Summary Of The Paper:**

This study uses YOLOv11 and Vision Transformer models to detect and count number of fragments in whole slide image datasets.

**Weaknesses:**

The models they use were developed by other researchers.
The comparison must be made to the similar model they mention in the intro which uses YOLOv5 and Fast R-CNN.

---

### Official Review · Reviewer_Xffj · 2025-07-19
**Automating Fragment Counting in Digital Pathology Review**

**Confidence:** 4
**Clarity Of Writing:** excellent
**Clinical Significance:** great
**Methodological Novelty:** great
**Overall Rating:** 8

**Experiments And Results:**

excellent

**Questions For The Authors:**

* None. This paper is easy to follow and is well structured.

**Strengths:**

* The paper directly defines the problem and propose a structured methodical solution
* Large dataset
* Analysis across multiple state of the art models was performed and clearly explained
* The proposed pipeline architecture yielded higher performance across the board
* Details were well explained and the paper was easy to follow
* Demonstration of multi-model pipeline shows promise in a wide area of application, outside of the papers scope
* Diagrams are well sized and easy to read/understand

**Summary Of The Paper:**

This paper highlights the difficulty and variability of manually counting tissue fragments on pathology slides, motivating the need for improved automated intervention. The authors evaluate state-of-the-art models on a robust multi-organ dataset and introduce a hybrid pipeline, HFC-Net, that achieves superior overall performance. They also apply auxiliary techniques (e.g., a rejection strategy) across models to further boost accuracy and precision. Finally, they compare human observer performance with HFC-Net, showing the proposed system consistently outperforms observers across all reported metrics.

**Weaknesses:**

* The HFC-Net needs a clearer and stronger introduction. This is the main novelty in the paper, yet that was not made clear when it was first introduced.
* Nothing else to add, very good job!

---

### Official Review · Reviewer_BTFx · 2025-07-22
**The paper presents a promising hybrid approach combining YOLOv11 and Vision Transformer for automatic fragment detection and counting in Digital Pathology. The model is evaluated on a diverse real-world dataset and compared against human performance and baseline models, demonstrating strong results. However, the manuscript would benefit from a more comprehensive review of related work, additional details on model training (e.g., hyperparameter tuning and data augmentation), and refinement of certain language choices to enhance clarity and formality.**

**Confidence:** 4
**Clarity Of Writing:** great
**Clinical Significance:** good
**Methodological Novelty:** good
**Overall Rating:** 6

**Experiments And Results:**

great

**Questions For The Authors:**

1 - The state-of-the-art part should be expanded to include additional relevant studies along with their reported results. This would enable a more comprehensive discussion and facilitate comparison between the proposed method and existing approaches in the literature.

2 - More details could be provided regarding the hyperparameter tuning process and the data augmentation strategies employed during model training.

3 - It is recommended to revise the manuscript to eliminate expressions typically associated with large language models, such as overly stylized phrases or extended dashes (e.g., “with no systematic comparison against expert manual counts—limiting conclusions about interobserver variability and clinical robustness”).

**Strengths:**

The main strengths of the paper include the use of a hybrid architecture that combines advanced models, as well as a thorough evaluation strategy that assesses performance at multiple levels: overall, by individual organ type, and through comparison with human performance.

**Summary Of The Paper:**

The manuscript introduces a hybrid architecture that integrates YOLOv11 and Vision Transformer (ViT) for the automated detection and counting of tissue fragments in Digital Pathology. To validate this approach, a comprehensive real-world dataset comprising tissue images from various organs is utilized. The performance of the proposed model is benchmarked against established reference models. Furthermore, the evaluation includes a breakdown by organ type and a comparison with human performance on the same task. The results demonstrate satisfactory outcomes, with the proposed method outperforming the reference models.

**Weaknesses:**

The main weakness of the article lies in the limited coverage of related work in the state of the art. The introduction references only one prior study, and it lacks details on the performance metrics reported in that work. As a result, despite the thorough internal evaluation and comparisons conducted, the absence of external benchmarking against existing models in the field limits the ability to fully assess the originality and novelty of the proposed approach.

---

### Official Review · Reviewer_Xuw8 · 2025-07-22
**Clear writing, but needing a few clarifications on experimental dataset**

**Confidence:** 4
**Clarity Of Writing:** great
**Clinical Significance:** good
**Methodological Novelty:** fair
**Overall Rating:** 5
**Final Rating:** 6

**Experiments And Results:**

good

**Questions For The Authors:**

- How many instances of each count are there in the dataset? While the description of each organ tissue is broken down, I do not recall if the distribution of the number of fragments was provided. I recall that the max count was set to 10+, but not the prevalence of each amount.

- Does CountPath named in the title of the paper reference something about the model or is it just referencing the topic of the slide segmentation?

**Strengths:**

- Clear writing and structure to the paper to flow through defining the problem, the methods, results, and discussion.

- Good dataset collected for the task of slides fragment counting and segmentation. Having multiple cell types and breaking out the results on each type improves the clinical significance of the dataset and the paper.

**Summary Of The Paper:**

This paper discusses the challenge of automating slide fragment counting whole-slide-images.  This task is performed to monitor and validate slides with tissue samples from various organs, which may contain various numbers of samples per set or sets per slide. The paper proposes a hybrid ViT and YOLO architecture to utilize the strong performance of ViT on set detection and YOLO for fragment counting in each set. The combined model achieved higher performance than any of the ViT or YOLO single models used as benchmarks, and showed generalizability across the varied tissue samples used in the custom dataset.

**Weaknesses:**

- Offering to make a public version of the dataset and code available would greatly improve the results by improving transparency and reproducibility of the results.

- Imbalance in training vs testing classes: I noticed that the colorectal samples comprise 60% of the training samples (~1500 of 2500), yet only 29% of the evaluation test set (200/701). Random sampling could cause some differences in class distribution, but I didn’t find anywhere describing how the train/validation/test split was created.  Explaining how the split was made or rebalancing the split by class would strengthen the experimental setup.